# Stripes in the extended $t - t'$ Hubbard model:
# A variational Monte Carlo analysis

**Vito Marino[1], Federico Becca[2] and Luca F. Tocchio[1]★**

**1** Institute for Condensed Matter Physics and Complex Systems, DISAT,
Politecnico di Torino, I-10129 Torino, Italy
**2** Dipartimento di Fisica, Università di Trieste, Strada Costiera 11, I-34151 Trieste, Italy

★ luca.tocchio@polito.it

## Abstract

By using variational quantum Monte Carlo techniques, we investigate the instauration of stripes (i.e., charge and spin inhomogeneities) in the Hubbard model on the square lattice at hole doping $\delta = 1/8$, with both nearest- ($t$) and next-nearest-neighbor hopping ($t'$). Stripes with different wavelengths $\lambda$ (denoting the periodicity of the charge inhomogeneity) and character (bond- or site-centered) are stabilized for sufficiently large values of the electron-electron interaction $U/t$. The general trend is that $\lambda$ increases going from negative to positive values of $t'/t$ and decreases by increasing $U/t$. In particular, the $\lambda = 8$ stripe obtained for $t' = 0$ and $U/t = 8$ [L.F. Tocchio, A. Montorsi, and F. Becca, SciPost Phys. 7, 21 (2019)] shrinks to $\lambda = 6$ for $U/t \gtrsim 10$. For $t'/t < 0$, the stripe with $\lambda = 5$ is found to be remarkably stable, while for $t'/t > 0$, stripes with wavelength $\lambda = 12$ and $\lambda = 16$ are also obtained. In all these cases, pair-pair correlations are highly suppressed with respect to the uniform state (obtained for large values of $|t'/t|$), suggesting that striped states are not superconducting at $\delta = 1/8$.



# 1 Introduction

Since the first experimental evidence of superconductivity in copper-oxide materials [1], high-temperature cuprate superconductors attracted a lot of interest, both for the potential applications and for the mysteries in their theoretical description [2,3]. Among many unsolved questions, there is the interplay between superconductivity and charge and spin inhomogeneities, which have been dubbed *stripes* [4]. Indeed, the possibility that cuprate superconductors may be on the verge of charge and/or spin instabilities has been suggested not long after their discovery [5,6]. A strong signature for spin inhomogeneities came from neutron scattering measurements in $La_{1.48}Nd_{0.4}Sr_{0.12}CuO_4$ [7]. Since then, a variety of experimental probes pointed to the presence of spin or charge orders [8–12]. Recently, charge-density fluctuations with short correlation lengths have been shown to be present in a wide region of the temperature-doping phase diagram [13,14]. Still, the implication of stripes in the insurgence of superconductivity is not clear. For example, $La_{2-x}Ba_xCuO_4$ exhibits a sharp depression in the superconducting transition temperature around $x = 1/8$, while $La_{2-x}Sr_xCuO_4$ shows only a small kink at the same electron density. The only difference between the two compounds is the structural phase transition that occurs at low temperatures when Ba is present [15–17]. The possibility that stripes and superconductivity are two sides of the same coin comes from the pair-density-wave scenario [18]. Here, a "mother state" has a superconducting order parameter that varies periodically in the material, in such a way that its spatial average vanishes. Then, these pair-density-wave states are unstable toward true long-range orders, e.g., charge- or spin-density waves or superconductivity. This fact could explain the rich phase diagram observed in cuprates (including the elusive pseudogap phase).

The simplest model that has been considered to reproduce the essential features of the cuprates' phase diagram is the single-band Hubbard model on the square lattice, which is characterized by the nearest-neighbor hopping $t$ and the on-site electron-electron repulsion $U$. Unfortunately, obtaining exact solutions or even accurate approximations for the ground state and for low-energy excitations is far from being trivial. In the recent past, several states have been proposed, being very close in energy, and different conclusions have been obtained by various numerical and analytical methods [19]. The search for the absolute energy minimum represents a formidable task, especially for the relevant electron densities that are expected to give rise to unconventional superconductivity. In this respect, it is important to assess the role of additional parameters, in enhancing the tendency towards superconductivity or stripes. For that, here we consider the extended Hubbard model that includes the next-nearest-neighbor hopping $t'$:

$$\mathcal{H} = -t \sum_{\langle R,R'\rangle,\sigma} c^\dagger_{R,\sigma} c_{R',\sigma} - t' \sum_{\langle\langle R,R'\rangle\rangle,\sigma} c^\dagger_{R,\sigma} c_{R',\sigma} + \text{H.c.} + U \sum_R n_{R,\uparrow} n_{R,\downarrow}, \tag{1}$$

where $c^\dagger_{R,\sigma}$ ($c_{R,\sigma}$) creates (destroys) an electron with spin $\sigma$ on site $R$ and $n_{R,\sigma} = c^\dagger_{R,\sigma} c_{R,\sigma}$ is the electron density per spin $\sigma$ on site $R$. In the following, we indicate the coordinates of the sites with $\mathbf{R} = (x, y)$. The nearest- and next-nearest-neighbor hoppings are denoted by $t$ and $t'$, respectively, while $U$ is the on-site Coulomb interaction. The electron density is given by $n = N/L$, where $N$ is the number of electrons and $L$ is the total number of sites. The hole doping is defined as $\delta = 1 - n$.

The presence of stripes in the Hubbard model has been first proposed by using density-matrix renormalization group (DMRG) on 6-leg ladders for the case with $t' = 0$ [20,21]. Then, further investigations supported the idea that charge and spin inhomogeneities may pervade the phase diagram of the Hubbard model [22,23]. In this context, a recent work [24], which combined a variety of numerical techniques, focused on the representative doping $\delta = 1/8$ and $U/t = 8$. Here, the existence of stripes with different lengths has been reported, the

lowest-energy stripe being a bond-centered one with periodicity $\lambda = 8$ in the charge sector and $2\lambda = 16$ in the spin sector. As a consequence, the enlarged unit cell of length $\lambda$ contains, on average, one hole, as obtained by previous Hartree-Fock calculations [25–28]. Still, different wavelengths (from $\lambda = 5$ to 8) were found very close in energy in Ref. [24]. Electron pairing was not found for $\lambda = 8$, but it was detected at other wavelengths. In addition, further studies, based on variational Monte Carlo [29], variational auxiliary-field Monte Carlo [30] and a combination of constrained-path auxiliary field Monte Carlo and DMRG [31], highlighted the absence of superconductivity at doping 1/8, the system being possibly an insulator.

The stabilization of stripes away from $\delta = 1/8$ has been investigated by cellular dynamical mean-field theory [32], variational Monte Carlo [29, 33] and auxiliary-field Monte Carlo [34]. Superconductivity has been proposed to coexist with stripes, with the remarkable exceptions of $\delta = 1/8$ and $\delta = 1/6$ [29]. Finally, recent calculations based on variational auxiliary-field Monte Carlo suggest that stripes exist in the low-doping regime, while a uniform superconducting state is stable in a sizable portion of the phase diagram for $\delta \approx 0.2$ [30]. Besides these works, it is important to investigate the effect of a next-nearest neighbor hopping, which is present in all the materials of the cuprate family [35]. Determinant [36] and variational [37] quantum Monte Carlo calculations suggested that the length of the stripe is reduced in presence of a negative next-nearest-neighbor hopping $t'$. In addition, an extensive study based on the projected entangled-pair states indicated that the half-filled stripe with $\lambda = 4$ is present for $-0.4 \lesssim t'/t \lesssim -0.1$ [38]. Concomitant superconductivity is not present at $\delta = 1/8$ and appears only for larger doping values. Conversely, on the 4-leg ladder, DMRG calculations reported a completely different picture, where the presence of the next-nearest neighbor hopping drives the system to have both superconducting and charge correlations that decay as a power law, while spin correlations decay exponentially [39, 40]. However, the 4-leg ladder may be a peculiar geometry, where, for example, ordinary $d$-wave pairing is replaced by a plaquette pairing [41].

In this work, we employ the variational Monte Carlo method with backflow correlations to investigate the effect of the next-nearest neighbor hopping $t'$ and of the Coulomb repulsion $U$ on the presence of stripes and superconductivity at doping 1/8. Our simulations are performed on ladder systems, with $L_y$ legs and $L_x$ rungs, the total number of sites being $L = L_x \times L_y$. In particular, we focus on a 6-leg cylinder geometry, that has been employed in DMRG calculations and is expected to capture the properties of truly two-dimensional clusters [24], while it allows us to accommodate long stripes along the rungs. We have also verified in selected cases that stripes are stable when working directly in two dimensions. Both bond-centered stripes (with even values of $\lambda$) and site-centered ones (with odd values of $\lambda$) are considered. We start by fixing $t' = 0$. Here, the ground state changes from a uniform metal (at low values of $U/t$) to an insulator with an optimal stripe of wavelength $\lambda = 8$ and eventually becomes a striped metal with $\lambda = 6$. Then, we analyze the effect of the next-nearest-neighbor hopping for two values of the electron-electron interaction $U/t = 8$ and 12. The general feature is that a negative (positive) ratio $t'/t$ tends to reduce (increase) the length of the stripe, until a uniform state is obtained for large values of $|t'/t|$ (with Néel order for $t'/t > 0$). In particular, for $t'/t < 0$, the (site-centered) stripe with $\lambda = 5$ dominates a large region of the phase diagram. Superconducting correlations are strongly suppressed when stripes are present, indicating that $\delta = 1/8$ is not superconducting for whatsoever stripe wavelengths. Our work is complementary to the variational Monte Carlo one of Ref. [37]. Indeed, besides a different definition of the variational wave functions, we considered different values of the Coulomb repulsion and the role of the sign of $t'/t$, which are not addressed in Ref. [37]. By contrast, they considered different dopings, while we focused only on doping $\delta = 1/8$.

## 2 Variational Monte Carlo method

Our numerical results are based upon the definition of suitable correlated variational wave functions, whose physical properties can be evaluated within Monte Carlo techniques [42]. In particular, electron-electron correlation is inserted by means of the so-called Jastrow factor [43, 44] on top of a Slater determinant or a Bardeen-Cooper-Schrieffer (BCS) state. In addition, backflow correlations, as described in Refs. [45, 46], are implemented. The latter ingredient is important to get accurate variational states, whose accuracy is comparable to other state-of-the-art numerical approaches [47]. Our simulations are performed on ladders with $L = L_x \times 6$ sites and periodic boundary conditions along both the $x$ and the $y$ directions. In order to fit charge and spin patterns in the cluster, we take $L_x = 2k\lambda$ (with $k$ integer).

The wave function is defined by:

$$|\Psi\rangle = \mathcal{J}_d |\Phi_0\rangle, \tag{2}$$

where $\mathcal{J}_d$ is the density-density Jastrow factor and $|\Phi_0\rangle$ is a state that is constructed from the ground state of an auxiliary noninteracting Hamiltonian by applying backflow correlations [45, 46].

The Jastrow factor is given by

$$\mathcal{J}_d = \exp\left(-\frac{1}{2}\sum_{R,R'} v_{R,R'} n_R n_{R'}\right), \tag{3}$$

where $n_R = \sum_\sigma n_{R,\sigma}$ is the electron density on site $R$ and $v_{R,R'}$ are pseudopotentials that are optimized for every independent distance $|\mathbf{R} - \mathbf{R}'|$ of the lattice.

The auxiliary noninteracting Hamiltonian includes different terms:

$$\mathcal{H}_{\text{aux}} = \mathcal{H}_0 + \mathcal{H}_{\text{charge}} + \mathcal{H}_{\text{spin}} + \mathcal{H}_{\text{AF}} + \mathcal{H}_{\text{BCS}}. \tag{4}$$

The first one defines the kinetic energy of the electrons:

$$\mathcal{H}_0 = -t \sum_{\langle R,R'\rangle,\sigma} c^\dagger_{R,\sigma} c_{R',\sigma} - \tilde{t}' \sum_{\langle\langle R,R'\rangle\rangle,\sigma} c^\dagger_{R,\sigma} c_{R',\sigma} + \text{H.c.}, \tag{5}$$

where the value of the nearest neighbor hopping parameter $t$ is fixed to be equal to the one in the Hubbard Hamiltonian of Eq. (1), in order to set the energy scale. Then, the second and third terms describe striped states that can be either bond-centered or site-centered:

$$\mathcal{H}_{\text{charge}} = \Delta_c \sum_R \cos[Q(x-x_0)]\left(c^\dagger_{R,\uparrow} c_{R,\uparrow} + c^\dagger_{R,\downarrow} c_{R,\downarrow}\right), \tag{6}$$

and

$$\mathcal{H}_{\text{spin}} = \Delta_s \sum_R (-1)^{x+y} \sin\left[\frac{Q}{2}(x-x_0)\right]\left(c^\dagger_{R,\uparrow} c_{R,\uparrow} - c^\dagger_{R,\downarrow} c_{R,\downarrow}\right), \tag{7}$$

where $x_0 = 1/2$ ($x_0 = 0$) for bond-centered (site-centered) stripes. In general, stripes can be either bond-centered or site-centered, with charge and spin modulations in the $x$ direction. The charge modulation has periodicity $\lambda = 2\pi/Q$ in all cases; instead, due to the $\pi$-phase shift in the antiferromagnetic order across the sites (site) with maximal hole density, the spin modulation is $2\lambda = 4\pi/Q$ when $\lambda$ is even and $2\pi/Q$ when $\lambda$ is odd. In all cases, Néel order is assumed for the spin modulation along the $y$ direction

In addition, a standard Néel order with pitch vector $\mathbf{K} = (\pi, \pi)$ can be considered, as the fourth term in Eq. (4):

$$\mathcal{H}_{\text{AF}} = \Delta_{\text{AF}} \sum_R (-1)^{x+y}\left(c^\dagger_{R,\uparrow} c_{R,\uparrow} - c^\dagger_{R,\downarrow} c_{R,\downarrow}\right). \tag{8}$$

Finally, the last term induces electron pairing:

$$\mathcal{H}_{\text{BCS}} = \sum_{R,\eta=x,y} \Delta_{R,R+\eta}(c^\dagger_{R,\uparrow}c^\dagger_{R+\eta,\downarrow} - c^\dagger_{R,\downarrow}c^\dagger_{R+\eta,\uparrow}) + \text{H.c.} - \mu \sum_{R,\sigma} c^\dagger_{R,\sigma}c_{R,\sigma}, \tag{9}$$

where the pairing amplitude may also be modulated in real space:

$$\Delta_{R,R+x} = \Delta_x \left|\cos\left[\frac{Q}{2}(x + \frac{1}{2} - x_0)\right]\right|, \qquad \Delta_{R,R+y} = -\Delta_y \left|\cos\left[\frac{Q}{2}(x - x_0)\right]\right|. \tag{10}$$

This modulation has been named "in phase" in Ref. [48]. A different kind of modulation, without the absolute values in Eq. 10, has been introduced in Ref. [48]; it is named "antiphase" and is characterized by a vanishing spatial average. Alternatively, a uniform pairing amplitude (with $d$-wave symmetry) can be considered, corresponding to $Q = 0$; a fictitious chemical potential $\mu$ is also included in $\mathcal{H}_{\text{BCS}}$.

The auxiliary Hamiltonian of Eq. (4) can be diagonalized by standard methods. Its ground state is then constructed. On top of it, backflow correlations are inserted to define $|\Phi_0\rangle$ of Eq. (2), following our previous works [45, 46].

The optimizations of the variational wave function are performed by imposing either $\Delta_{\text{AF}} = 0$, fixing a given stripe wavelength $\lambda$, and optimizing $\Delta_x$, $\Delta_y$, $\mu$, $\Delta_c$, $\Delta_s$, and $\tilde{t}'$ (as well as all the pseudopotentials in the Jastrow factor and the backflow parameters) or imposing $\Delta_c = \Delta_s = 0$ and a uniform pairing, optimizing all the other parameters. In the former case, we will denote the resulting wave function as "striped state"; in the latter one we will denote it as "uniform state" (even if Néel order can be present, whenever $\Delta_{\text{AF}} \neq 0$). We have also tried to optimize the "antiphase" pairing, while setting $\Delta_c = \Delta_s = \Delta_{\text{AF}} = 0$, since this wave function gives a possible parametrization of a pair-density-wave state. However, we have found that this option gives a higher energy with respect to the uniform one, both at $t' = 0$ and at $t'/t < 0$.

In order to unveil the presence of charge and spin inhomogeneities, we compute static structure factors:

$$N(\mathbf{q}) = \frac{1}{L} \sum_{R,R'} \langle n_R n_{R'} \rangle e^{i\mathbf{q}\cdot(\mathbf{R}-\mathbf{R}')}, \tag{11}$$

and

$$S(\mathbf{q}) = \frac{1}{L} \sum_{R,R'} \langle S^z_R S^z_{R'} \rangle e^{i\mathbf{q}\cdot(\mathbf{R}-\mathbf{R}')}, \tag{12}$$

where $\langle\ldots\rangle$ indicates the expectation value over the variational wave function and $S^z_R = 1/2(c^\dagger_{R,\uparrow}c_{R,\uparrow} - c^\dagger_{R,\downarrow}c_{R,\downarrow})$. The presence of a peak (diverging in the thermodynamic limit) at a given $\mathbf{q}$ vector denotes the presence of charge order in the system. Moreover, the small-$q$ behavior of $N(\mathbf{q})$ allows us to assess the metallic or insulating nature of the ground state. Indeed, charge excitations are gapless when $N(\mathbf{q}) \propto |\mathbf{q}|$ for $|\mathbf{q}| \to 0$, while a charge gap is present whenever $N(\mathbf{q}) \propto |\mathbf{q}|^2$ for $|\mathbf{q}| \to 0$ [46, 49].

The possible existence of superconductivity is investigated by computing correlation functions between Cooper pairs at distance $r$. In particular, we can consider pairs along the $y$ direction, so that:

$$D(r) = \langle \left(c^\dagger_{R,\uparrow}c^\dagger_{R+y,\downarrow} - c^\dagger_{R,\downarrow}c^\dagger_{R+y,\uparrow}\right)\left(c^\dagger_{R',\uparrow}c^\dagger_{R'+y,\downarrow} - c^\dagger_{R',\downarrow}c^\dagger_{R'+y,\uparrow}\right)\rangle, \tag{13}$$

where $R' = R + rx$.

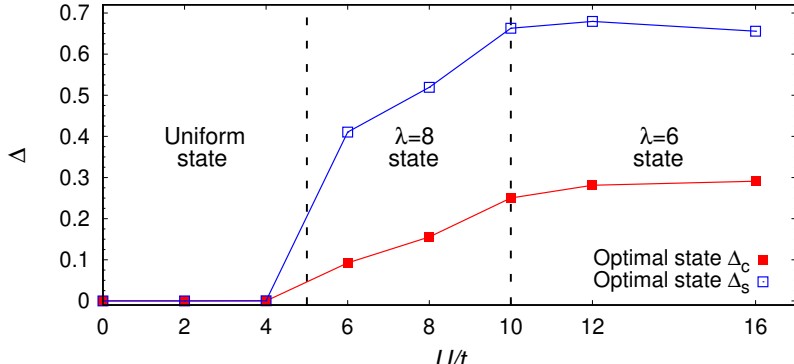

Figure 1: The variational parameters $\Delta_c$ (red full squares) and $\Delta_s$ (blue empty squares), for charge and spin modulations, see Eqs. (6) and (7). Results are shown for $t' = 0$, as a function of $U/t$. Data with $\lambda = 8$ and for the uniform case are reported for $L_x = 16$, while data with $\lambda = 6$ are shown for $L_x = 24$. At $U/t = 10$, where two solutions are degenerate, we show data for $\lambda = 6$.

Table 1: Energies per site (in unit of $t$) of the stripe state with $\lambda = 8$ and of the uniform state, for $U/t = 8$ and $t' = 0$, for different values of the lattice size $L = L_x \times 6$. The error bar on the energy is always smaller than $10^{-4}$.

| $L_x$ | Energy striped state ($\lambda = 8$) | Energy uniform state |
|---|---|---|
| 16 | -0.7486 | -0.7439 |
| 32 | -0.7483 | -0.7437 |
| 48 | -0.7482 | -0.7435 |
| 64 | -0.7482 | -0.7435 |

## 3 Results

Here, we work out the optimal wavelength of the stripe, assessing its conducting or insulating nature, for the hole doping $\delta = 1/8$, either fixing $t' = 0$ and varying the strength of the electron-electron interaction $U$ or fixing $U$ and varying the next-nearest-neighbor hopping $t'$. In the latter case, the values $U/t = 8$ and $12$ are considered. The possible insurgence of superconductivity, also coexisting with stripes, is finally investigated. We have verified that, for even values of $\lambda$, site-centered and bond-centered stripes have the same energy (within the error bar), with the only exception of $\lambda = 4$. In this case, the site-centered stripe has slightly lower energy than the bond-centered one (the difference being $10^{-3} t$).

### 3.1 Varying $U/t$ with $t' = 0$

First of all, we analyze the case with $t' = 0$ and different values of $U/t$. We consider the variational energies for different stripe wavelengths $\lambda$, comparing them with the energy of a uniform state (which may still possess Néel order). For $U/t \lesssim 4$, all the striped wave functions are not stable, converging to the uniform state with vanishing parameters $\Delta_c$ and $\Delta_s$. In addition, neither Néel order nor electron pairing is obtained in the auxiliary Hamiltonian of Eq. (4), indicating that the best state is a standard metal. For $6 \lesssim U/t \lesssim 8$, the best energy is obtained for $\lambda = 8$; instead, for $U/t \gtrsim 10$ (we did not consider values of the Coulomb interaction larger than $U/t = 16$) the optimal stripe has $\lambda = 6$. For $U/t \approx 10$, the energies

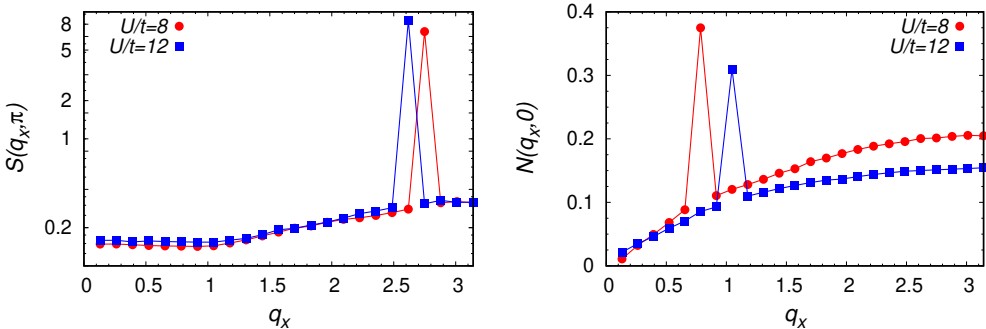

Figure 2: Left panel: Spin-spin correlation function $S(\mathbf{q})$ on a semi-log scale, as a function of $q_x$ with $q_y = \pi$. Results are reported for $t' = 0$ and $L_x = 48$. The best variational state has $\lambda = 8$ at $U/t = 8$ (red circles) and $\lambda = 6$ at $U/t = 12$ (blue squares). Right panel: Same as in the left panel, but on a linear scale for the static structure factor $N(\mathbf{q})$ with $q_y = 0$.

of these two states are very close to each other. The variational parameters $\Delta_c$ and $\Delta_s$ for the optimal state are reported in Fig. 1, as a function of $U/t$. They are both finite where stripes are present, with the spin modulation being stronger than the charge one, while they collapse to zero for $U/t \lesssim 4$.

Then, we briefly discuss the dependence of the variational energies on the lattice size. In particular, we focus our attention on $U/t = 8$, where the lowest-energy state has a stripe with $\lambda = 8$. The variational energies of the striped wave function and the uniform one are reported in Table 1, for lattice sizes ranging from $L_x = 16$ to $L_x = 64$. The energies have only tiny variations going from small to large systems, indicating a very fast convergence to the limit $L_x \to \infty$. Therefore, the optimal state can be obtained by comparing energies already on small sizes. By contrast, large clusters are necessary to discriminate between metallic and insulating properties, since this requires the evaluation of the small-$q$ behavior of $N(\mathbf{q})$.

The actual presence of charge and spin order in the wave function can be directly detected in the static structure factors of Eqs. (11) and (12), as shown in Fig. 2. Here, clear peaks, diverging with the system size (not shown), are present, for $\mathbf{q} = (\frac{2\pi}{\lambda}, 0)$ in the charge correlations $N(\mathbf{q})$ and for $\mathbf{q} = (\pi(1 - \frac{1}{\lambda}), \pi)$ in the spin correlations $S(\mathbf{q})$. In Ref. [29], two of us suggested that stripes are driven by spins rather than charges. Then, the shortening of the stripe when increasing the electron-electron repulsion could be explained by noticing that the antiferromagnetic energy scale $J = \frac{4t^2}{U}$ increases when decreasing $U/t$, as long as the system remains sufficiently correlated (i.e., $U/t \gtrsim 4$), and that for longer stripes the number of nearest-neighbor antiparallel spins increases.

Let us now assess the metallic or insulating behavior of the optimal state, as it can be extracted from the small-$q$ behavior of the static structure factor $N(\mathbf{q})$. In Fig. 3, we show $N(\mathbf{q})/q_x$ for the striped states at $U/t = 8$ and at $U/t = 12$. We observe that for $U/t = 8$ the behavior of $N(\mathbf{q})/q_x$ extrapolates to zero when the lattice size increases, compatibly with an insulating behavior. On the contrary, for $U/t = 12$, $N(\mathbf{q}) \approx q_x$ at small $q_x$, clearly indicating that the state is metallic. The latter result can be explained by the fact that the wavelength of the stripe is not commensurate with the doping, since an integer number of holes cannot be accommodated in the unit cell of the charge modulation.

Finally, we report that small superconducting parameters $\Delta_x$ and $\Delta_y$ can be stabilized in the auxiliary Hamiltonian of Eq. (10) for $U/t \gtrsim 4$. However, they do not give rise to sizable pair-pair correlations $D(r)$, as defined in Eq. (13), see Fig. 4. Both the optimal striped state with $\lambda = 8$ at $U/t = 8$ and the one with $\lambda = 6$ at $U/t = 12$ show strongly suppressed pair-

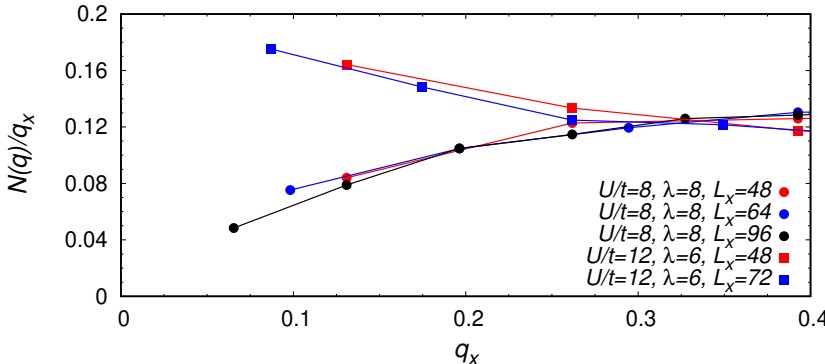

Figure 3: Static structure factor (divided by $q_x$) $N(\mathbf{q})/q_x$ as a function of $q_x$ with $q_y = 0$. Data are reported for $t' = 0$ and different values of $L_x$. The optimal state has $\lambda = 8$ for $U/t = 8$ (circles) and $\lambda = 6$ for $U/t = 12$ (squares).

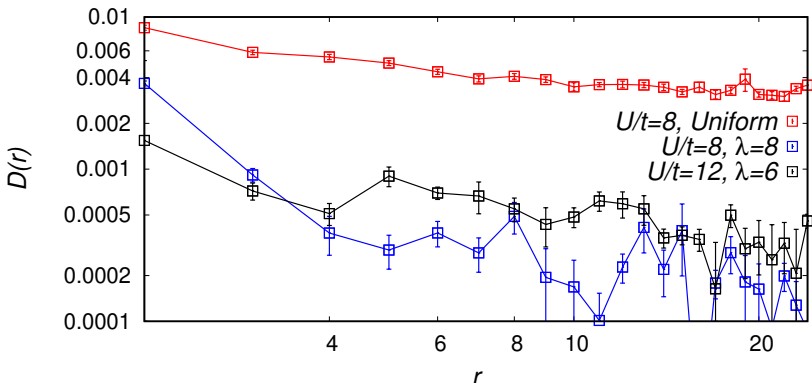

Figure 4: Pair-pair correlations $D(r)$ as a function of $r$ for the optimal striped state with $\lambda = 8$ at $U/t = 8$ (blue squares), the one with $\lambda = 6$ at $U/t = 12$ (black squares), and the uniform state (not optimal) at $U/t = 8$ (red squares). Data are shown for $L_x = 48$ on a log-log scale.

pair correlations with respect to the uniform case at $U/t = 8$ (that has a higher variational energy). Remarkably, the two striped states have a similar suppression in $D(r)$ even if one state is insulating and the other one is metallic. We mention that results for the striped states are shown for modulated pairings in the variational wave function, see Eq. (10), but the results are similar also imposing a uniform pairing.

## 3.2 Varying $t'/t$ with $U/t = 8$ and $12$

We fix now the value of $U/t$ and vary $t'/t$. We start by considering $U/t = 8$, see Fig. 5. Here, both $\Delta_c$ and $\Delta_s$ are finite in a relatively large regime, i.e., for $-0.45 \lesssim t'/t \lesssim 0.25$. For $t'/t < 0$, the stripe wavelength reduces to $\lambda = 6$ and then to $\lambda = 5$, that remains the optimal state in a large range of $t'/t$, until the optimal state becomes uniform, with no magnetic Néel order, for $t'/t \lesssim -0.5$. The transition is first order, with a clear jump in the values of $\Delta_c$ and $\Delta_s$. The maximum energy gain between the striped state and the uniform one is reached for $t'/t \approx -0.25$, which is often considered to be a prototypical value for the cuprate family [35]. In this case, we have verified that the stripe with $\lambda = 5$ represents the best-energy solution also in two dimensional systems, e.g., on the $L = 20 \times 20$ cluster. For $t'/t > 0$, the optimal $\lambda$ increases first to $\lambda = 12$ and then to $\lambda = 16$, until the system becomes uniform again, but

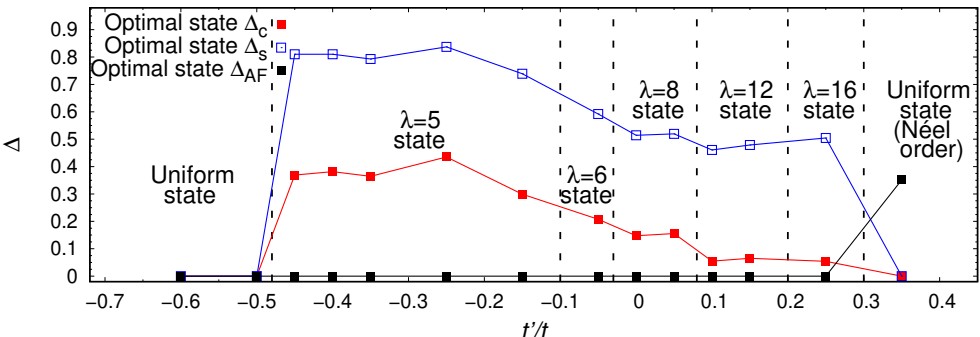

Figure 5: The variational parameters $\Delta_c$ (red full squares), $\Delta_s$ (blue empty squares), and $\Delta_{AF}$ (black full squares), see Eqs. (6), (7), and (8). Results are shown for $U/t = 8$ as a function of $t'/t$. Data with $\lambda = 8$ are reported for $L_x = 16$, with $\lambda = 6$ and 12 for $L_x = 24$, with $\lambda = 16$ for $L_x = 32$, and with $\lambda = 5$ for $L_x = 40$; finally, data for the uniform cases are reported for $L_x = 16$.

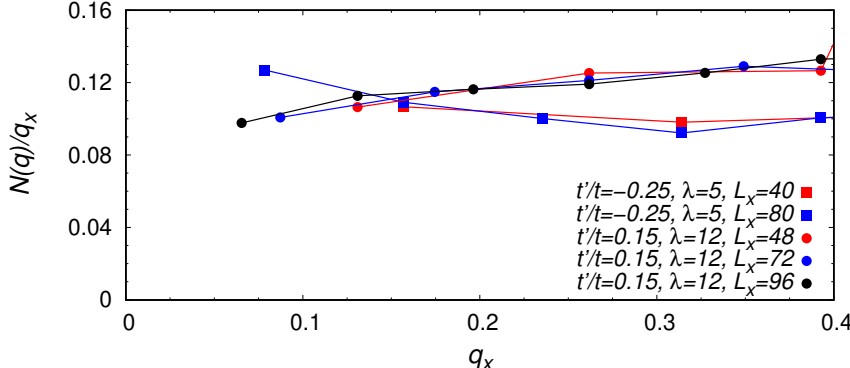

Figure 6: Static structure factor (divided by $q_x$) $N(\mathbf{q})/q_x$ as a function of $q_x$ with $q_y = 0$. Results are reported for $U/t = 8$ at $t'/t = -0.25$ for the optimal striped state with $\lambda = 5$ (squares) and at $t'/t = 0.15$ for the optimal striped state with $\lambda = 12$ (circles). Different values of $L_x$ are shown.

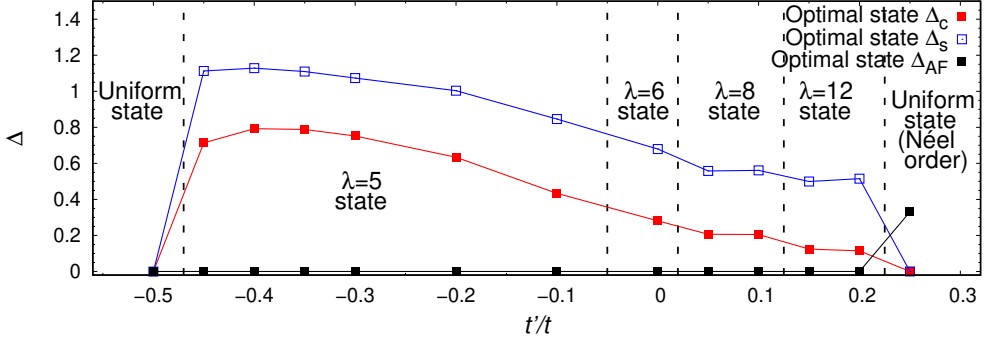

Figure 7: The same as in Fig. 5 but for $U/t = 12$. All data are shown for $L_x = 48$, except data for $\lambda = 5$ for which $L_x = 40$. At $t'/t = 0.25$, where two solutions are degenerate, we show data for the uniform state with Néel order.

with antiferromagnetic Néel order. In this case, the transition is weakly first order. In general, a large value of $|t'/t|$ frustrates the stripe pattern, with a striking difference between negative and positive values of $t'/t$; while the former one gives rise to a uniform ground state, the latter one stabilizes a relatively strong Néel order (for very large values of $t'/t \approx 0.75$, we expect that Néel order is replaced by a collinear order with pitch vector $\mathbf{K} = (\pi, 0)$ or $(0, \pi)$, as in the half-filled case with $\delta = 0$ [45]). The stability of Néel order up to large values of doping in the hole doped Hubbard model with $t'/t > 0$ (or equivalently in the electron doped Hubbard model with $t'/t < 0$) has been discussed, for example, in Ref [50]. We remark that we cannot exclude the presence of longer stripes for $t'/t > 0$: their detection would require very large clusters, being also difficult to distinguish them energetically from the uniform solution with Néel order.

In Fig. 6, we show the behavior of $N(\mathbf{q})/q_x$ for the optimal states with $\lambda = 5$ at $t'/t = -0.25$ and for $\lambda = 12$ at $t'/t = 0.15$. Here, the results of the striped states are consistent with a metallic behavior, indicating again that a striped state is insulating only for particular values of its wavelengths, e.g., for $\lambda = 8$ at $\delta = 1/8$.

Finally, we want to assess the properties for a larger value of the electron-electron interaction, i.e., $U/t = 12$. The reason for doing this analysis is to investigate the effect of correlations and to verify whether stripes with $\lambda = 4$ (recently obtained within tensor-network methods for $U/t = 10$ [38]) may be stabilized or not. Actually, this is not the case since the stripe with $\lambda = 5$ continues to dominate the phase diagram for $t'/t < 0$, see Fig. 7, where we report the variational parameters $\Delta_c$ and $\Delta_s$ for the optimal state, as a function of $t'/t$. Nevertheless, the stripe with $\lambda = 4$ is quite close in energy: for example, at $t'/t = -0.4$, the energy per site for the state with $\lambda = 4$ is $E/t = -0.6215$, while the one with $\lambda = 5$ is $E/t = -0.6220$. In Table 2, we also show the energy of the best striped state and of the uniform one, as a function of $t'/t$. As observed also for $U/t = 8$, for $t'/t < 0$, the optimal stripe wavelength decreases until a first-order transition to the uniform state takes place at $t'/t \approx -0.5$. Instead, for $t'/t > 0$, the stripe wavelength increases until the system becomes uniform (with Néel order) at $t'/t \approx 0.25$.

Table 2: Energy per site (in unit of $t$) for the best striped state $E_{\text{stripe}}$ and the uniform one $E_{\text{uniform}}$ (as well as their difference $\Delta E = E_{\text{stripe}} - E_{\text{uniform}}$), for $U/t = 12$, as a function of $t'/t$. Data are shown for $L_x = 48$, except for the case with $\lambda = 5$ for which $L_x = 40$. The error bar on the energy is always smaller than $10^{-4}$.

| $t'/t$ | $E_{\text{stripe}}$ | $E_{\text{uniform}}$ | $\Delta E$ |
|---|---|---|---|
| -0.5 | -0.6260 ($\lambda = 5$) | -0.6261 | 0.0001 |
| -0.45 | -0.6240 ($\lambda = 5$) | -0.6202 | -0.0038 |
| -0.4 | -0.6220 ($\lambda = 5$) | -0.6162 | -0.0058 |
| -0.35 | -0.6202 ($\lambda = 5$) | -0.6135 | -0.0067 |
| -0.3 | -0.6190 ($\lambda = 5$) | -0.6118 | -0.0072 |
| -0.2 | -0.6187 ($\lambda = 5$) | -0.6113 | -0.0074 |
| -0.1 | -0.6206 ($\lambda = 5$) | -0.6147 | -0.0059 |
| 0 | -0.6255 ($\lambda = 6$) | -0.6227 | -0.0028 |
| 0.5 | -0.6309 ($\lambda = 8$) | -0.6283 | -0.0026 |
| 0.1 | -0.6375 ($\lambda = 8$) | -0.6355 | -0.0020 |
| 0.15 | -0.6464 ($\lambda = 12$) | -0.6447 | -0.0017 |
| 0.20 | -0.6562 ($\lambda = 12$) | -0.6553 | -0.0009 |
| 0.25 | -0.6669 ($\lambda = 12$) | -0.6669 | 0 |

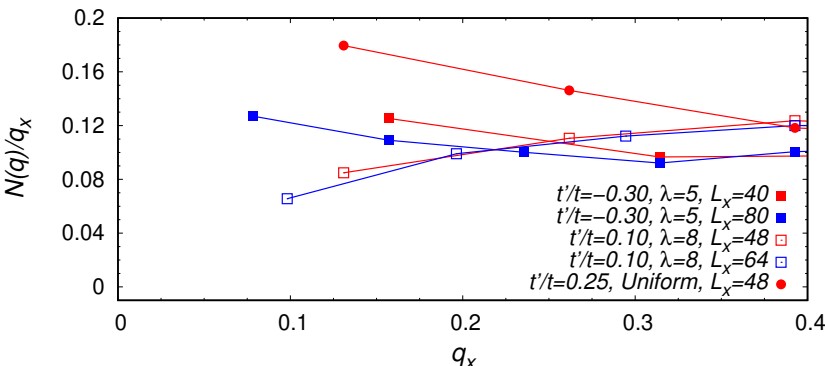

Figure 8: Static structure factor (divided by $q_x$) $N(\mathbf{q})/q_x$ as a function of $q_x$ with $q_y = 0$. Results are reported for $U/t = 12$ at $t'/t = -0.3$ for the optimal striped state with $\lambda = 5$ (full squares), at $t'/t = 0.1$ for the optimal striped state with $\lambda = 8$ (empty squares), and at $t'/t = 0.25$ for the uniform state with Néel order (full circles). Different values of $L_x$ are shown.

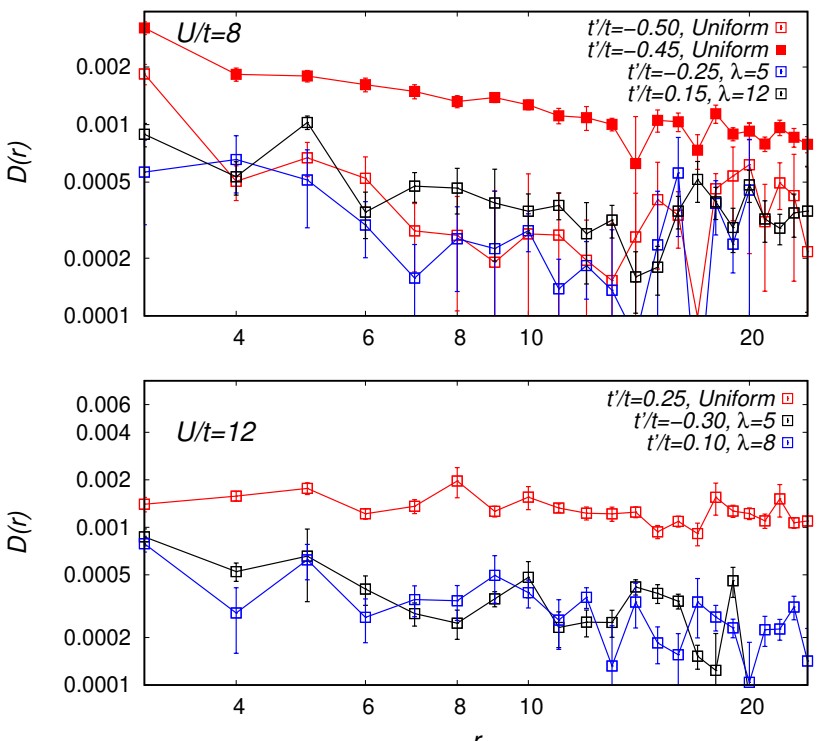

Figure 9: Pair-pair correlations $D(r)$ as a function of $r$. Lower panel: three optimal states at $U/t = 12$: the striped state with $\lambda = 5$ at $t'/t = -0.3$ (black squares), the striped state with $\lambda = 8$ at $t'/t = 0.1$ (blue squares), and the uniform state (with Néel order) at $t'/t = 0.25$ (red squares). Upper panel: three optimal states at $U/t = 8$: the striped state with $\lambda = 5$ at $t'/t = -0.25$ (blue squares), the striped state with $\lambda = 12$ at $t'/t = 0.15$ (black squares), and the uniform state at $t'/t = -0.5$ (red empty squares), together with the (non optimal) uniform state at $t'/t = -0.45$ (red full squares). Data are shown for $L_x = 48$, except data for $\lambda = 5$ that are shown at $L_x = 40$, on a log-log scale.

In Fig. 8, we show the behavior of $N(\mathbf{q})/q_x$ for three optimal states, two striped ones and the uniform state with Néel order. Again, these results support the idea that the only insulating stripe is the one with $\lambda = 8$, which is commensurate with the doping level. In this case, $N(\mathbf{q})/q_x$ extrapolates to zero as the lattice size increases. On the contrary, in the other two cases $N(\mathbf{q})/q_x$ extrapolates to a finite value, indicating that the state is metallic.

Also for finite values of $t'/t$, small superconducting parameters $\Delta_x$ and $\Delta_y$ can be stabilized in the auxiliary Hamiltonian of Eq. (10). However, as in the $t' = 0$ case, stripes do not coexist with sizable pair-pair correlations $D(r)$, see Fig. 9. The uniform state, both at positive and negative values of $t'/t$ (with Néel order in the former case) displays superconducting correlations, which, however, are progressively suppressed, with respect to the uniform case at $t' = 0$, when the ratio $|t'/t|$ increases.

## 4  Conclusions

By means of the variational Monte Carlo method, we have investigated the stability of stripes in the Hubbard model at doping $\delta = 1/8$, as a function of the next-nearest neighbor hopping $t'$ and of the Coulomb repulsion $U$. In particular, we compared the uniform state (which possibly includes electron pairing and Néel order) with striped states with various wavelengths $\lambda$. The summary of our results is shown in Fig. 10. The present work extends our previous analysis [29], where we observed an insulating filled stripe with $\lambda = 8$ at doping $\delta = 1/8$, with $t' = 0$ and $U/t = 8$.

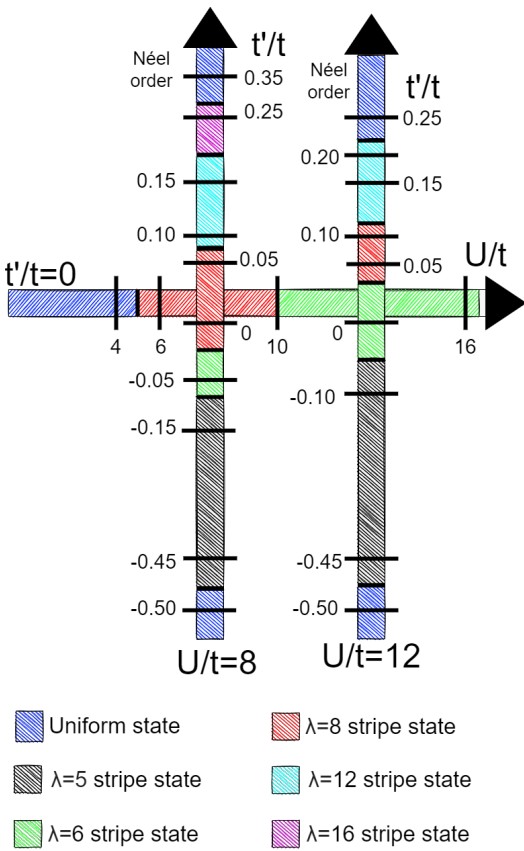

Figure 10: Schematic phase diagram as a function of $U/t$ for $t' = 0$ and as a function of $t'/t$ for $U/t = 8$ and $U/t = 12$.

Firstly, we have shown that a sufficiently large Coulomb interaction is necessary to stabilize the stripes, since a weakly correlated metallic phase is present for $U/t \lesssim 4$. Instead, for $U/t \gtrsim 4$, the phase diagram is pervaded by striped states, whose wavelength depends upon the value of $U/t$. In general, we found a shortening of the stripe length when increasing $U/t$. This result may be associated with the corresponding weakening of the antiferromagnetic energy scale, since a longer stripe is closer to the standard antiferromagnetic Néel order, which maximally satisfies the antiferromagnetic energy scale.

Then, we observed that stripes are shorter for negative values of $t'/t$ and become longer for positive values of it, as shown in Fig. 10, with the maximal energy gain between the striped case and the uniform one that is reached at $t'/t \approx -0.25$. We remark that stripes are present as long as the ratio $|t'/t|$ is not too large. The uniform ground state obtained for large enough $|t'/t|$ possesses Néel order for positive values of the ratio $t'/t$, while it has no spin order for negative values of it.

Finally, we report that only the stripe with $\lambda = 8$ is insulating, since it is the only one that is commensurate with the hole doping. All the other striped states are metallic. Remarkably, despite their metallic nature, stripes seem do not ever coexist with superconductivity at doping $\delta = 1/8$, since pair-pair correlations are always strongly suppressed with respect to the uniform superconducting state.

# Acknowledgments

We thank M. Grilli, R. Tateo, and A. Montorsi for useful discussions. Computational resources were provided by HPC@POLITO (http://www.hpc.polito.it).

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
