# Peer review of "Stripes in the extended $t-t^\prime$ Hubbard model: A Variational Monte Carlo analysis"

_SciPost Physics, doi:SciPost Phys. 12, 180 (2022)_

## Round 2 · Referee Report · Anonymous (Referee 1) · 2021-12-30

Report

Hubbard model is one of the fundamental Hamiltonians for strongly-correlated systems. It is also one of the minimal Hamiltonians for cuprate high-Tc superconductors. Although the form of the Hubbard model is simple, its ground state property is still highly controversial.

Recent rapid improvement in computational resources and numerical techniques has enabled intensive research on the ground-state property of the Hubbard model. In particular, stripe states gather much attention as a candidate for the ground state.

In this paper, the authors investigate the stability of stripe states in the Hubbard model on the square lattice at 1/8 hole doping. The authors apply the variational Monte Carlo method using Jastrow and backflow correlation factors.

First, the authors study the U dependence at t’=0. Stripe states are stabilized for U>4, and a period-8 (period-6) stripe state is realized in the intermediate U (large U) region. Superconductivity is suppressed in the stripe states.

Then, the authors investigate t’ dependence for U=8 and 12. For U=8, they obtain period-6, 8, 12, 16 stripes with the period increasing as t’ increases. For U=12, period-4, 6, 8 states are stabilized. Again the period increases with increasing t’. The superconducting correlation function is suppressed in the stripe states as in the cases of t’=0.

The paper is clearly written, and the present work is one of the important pieces of recent intensive numerical investigation of the Hubbard model. Thus, I recommend that this paper be accepted for publication in SciPost.

Below, I list several comments.

  1. The authors employ the backflow correlation factor, which is an off-diagonal correlation factor in Fock space. Therefore, it is expected to describe nontrivial correlation effects. Then, it would be interesting to investigate the effects of the backflow factor. In what kind of solution does the backflow factor become important? In other words, how much is the energy gain due to the backflow correlation factor? Is there a significant difference in energy gain between uniform and stripe states?

  2. Is it meaningful to put optimized Delta_x and Delta_y values in Figs. 5 and 7? Are the uniform states superconducting in the entire region (the authors briefly discuss superconductivity in Fig. 9, but the parameter region is limited.)?

  3. For general readers, I recommend the authors put discussions on the relevance to the pair-density-wave (PDW) scenario in the cuprate phase diagram.

Minor comments

  1. In Table II, it would be helpful to show the energy difference between uniform and stripe states.

  2. Throughout the paper, I recommend the authors make the lines thicker and the colors darker in the figures. The symbols are not very visible when printed.

  • validity: -
  • significance: -
  • originality: -
  • clarity: -
  • formatting: -
  • grammar: -

Author:  Luca Fausto Tocchio  on 2022-04-21  [id 2406]

(in reply to Report 1 on 2021-12-30)

We thank the referee for having carefully read our paper and for recommending it for publication. We provide below an answer to the referee's criticisms and comments.

Q: The authors employ the backflow correlation factor, which is an off-diagonal correlation factor in Fock space. Therefore, it is expected to describe nontrivial correlation effects. Then, it would be interesting to investigate the effects of the backflow factor. In what kind of solution does the backflow factor become important? In other words, how much is the energy gain due to the backflow correlation factor? Is there a significant difference in energy gain between uniform and stripe states?

A: The energy gain due to backfkow is particularly relevant for uniform variational wave functions, including superconducting ones. Indeed, the effect of backflow is to generate the electron correlations that are induced by the Hubbard-$U$, but not captured by the Jastrow-Slater parametrization based upon the auxiliary Hamiltonian of Eq.(4). In presence of charge and/or spin order, most of the electron correlation is described by including symmetry breaking terms in the auxiliary Hamiltonian, and, therefore, backflow terms are not crucial. By contrast, in absence of charge and/or spin order, the uniform state constructed from Eq.(4) misses a large part of the electron correlation due to the on-site Coulomb repulsion. To be more quantitative, some results can be already found in Fig. 2 of our previous publication [Tocchio, Montorsi, Becca, SciPost Phys. 7, 21 (2019)]. Some other data are shown in the table below. While the absence of backflow does not change the periodicity of the optimal stripe, the energy gain of the stripe solution with respect to the uniform one is clearly increased in the absence of backflow correlations:

$U/t$ $t'/t$ E(best striped state)-E(uniform state) (With Backflow) E(best striped state)-E(uniform state) (Without Backflow) 12 -0.4 -0.0045 -0.0282 8 -0.25 -0.0065 -0.0119 8 0.15 -0.0027 -0.0053

Interestingly, while in the presence of backflow correlations no stripe states can be stabilized, after optimization, at $U/t=4$ and $t'=0$, in the absence of backflow correlations the best solution is a stripe of $\lambda=8$, with an energy gain of -0.0024 with respect to the uniform state. All data are reported for a $L=48 \times 6$ lattice size.

Q: Is it meaningful to put optimized Delta_x and Delta_y values in Figs. 5 and 7? Are the uniform states superconducting in the entire region (the authors briefly discuss superconductivity in Fig. 9, but the parameter region is limited.)?

A: In the presence of stripes, a small value of the superconducting parameters $\Delta_x$ and $\Delta_y$ can be stabilized after optimization. This does not necessarily lead to the development of superconductivity, since correlations generated by Jastrow and backflow terms may suppress it. Indeed, pair-pair correlations are never sizable in the presence of stripes, as shown in the paper. For this reason, we decided not to show optimized $\Delta_x$ and $\Delta_y$ values in the paper. Instead, we confirm that uniform states are always superconducting, even if in the manuscript we have shown only selected cases. However, the superconductive order parameter (intended as the square root of the limit of the pair-pair correlations at large distances) is maximal at $t'=0$ and decreases for increasing $|t'/t|$, until it vanishes for large enough values of $|t'/t|$, consistently with what shown in the manuscript.

Q: For general readers, I recommend the authors put discussions on the relevance to the pair-density-wave (PDW) scenario in the cuprate phase diagram.

A: We thank the referee for having suggested to include a comment on the PDW scenario in the introduction, including the new reference by Agterberg et al., Annual Review of Condensed Matter Physics 11, 231 (2020). Within the PDW scenario, different ordered phases may emerge at low temperature, including stripes. A possible parametrization of the PDW state consists in taking a modulated pairing. As a check, we have then computed the variational energy associated to a state in which pairing is modulated and possess an "antiphase'' periodicity, as described in [A. Himeda, T. Kato, and M. Ogata, PRL 88, 117001 (2002)]. Our results show that this state is not competitive as a ground-state candidate, with respect to uniform superconductivity and stripes. We added a comment on this point in the manuscript.

Minor comments

Q: In Table II, it would be helpful to show the energy difference between uniform and stripe states.

A: We have followed the suggestion of the referee, adding one column to Table II.

Q: Throughout the paper, I recommend the authors make the lines thicker and the colors darker in the figures. The symbols are not very visible when printed.

A: We have made the lines thicker, following the suggestion of the referee. We hope that this will improve the visibility of the printed figures.

Author:  Luca Fausto Tocchio  on 2022-04-23  [id 2410]

(in reply to Luca Fausto Tocchio on 2022-04-21 [id 2406])

The table of the energy differences, with and without backflow is not very visible in the reply. We apologize for the inconvenience and we attach a properly formatted table to this message.

Attachment:

Energies_backflow.pdf

---

## Round 2 · Referee Report · Anonymous (Referee 2) · 2022-1-4

Report

In this paper the authors study the extended doped Hubbard model for U/t = 8 and U/t=12 at hole doping 1/8 for different values of the next-nearest neighbor hopping t' with variational Monte Carlo, based on fermionic wave functions including a density-density Jastrow factor and backflow correlations. Their auxiliary non-interacting Hamiltonian includes charge and spin modulations of a bond-centered stripe of wavelength lambda as well as an antiferromagnetic and pairing contribution, besides the regular kinetic term. Their main results include that lambda increases with increasing t'/t and decreasing U/t (in agreement with previous studies), that the stripes are not superconducting, and that uniform states are stabilized at sufficiently large values of |t'/t|. A lambda=4 stripe is found over a certain negative t'/t range for U/t=12, whereas for U/t=8 it is absent.

This work, which builds upon a previous work by two of the authors for t'=0, provides an interesting contribution to the ongoing efforts in understanding the phase diagram of the Hubbard model and the competition between uniform and stripe states. It provides further support for the shift in stripe period as a function of t'/t and U/t, and of the existence of a uniform superconducting state at large values of |t'/t|. Whether stripes coexist with superconductivity (in particular away from one hole per unit length) is still controversial, thus the absence of superconductivity in all stripes is an important and interesting result.

While I believe this work will eventually be suitable for publication in SciPost, it has some weak points, listed below, which should be addressed and improved first.

Requested changes

(1) It seems the authors have missed the fact that an extensive VMC study on the t-t' Hubbard model was already performed in [K. Ido, T. Ohgoe and M. Imada, PRB 97, 045138 (2018)] (however, without including backflow correlations). The authors cite this paper in their introduction (Ref. [35]), however, as a QMC calculation, and not as a VMC study. It would be important to compare their findings with these previous results in their discussion.

(2) The authors consider bond-centered stripes in their study. While there is a rather strong consensus that the lambda=8 stripe at 1/8 doping is bond-centered, this is not necessarily true for other stripe periods (or other dopings); in previous works there is rather a tendency that site-centered stripes are stabilized. Have the authors checked that site-centered stripes have a higher variational energy than bond-centered stripes? Or are the energies similar? It would be important to verify and discuss this point.

(3) It is not clear to me whether the authors have also considered odd values of lambda, since they only present results for even values. Have the authors checked the variational energies of odd lambda values and found that they are higher? Or was there a particular reason to restrict the study to even values? This point should be discussed/mentioned in the paper.

  • validity: -
  • significance: -
  • originality: -
  • clarity: -
  • formatting: -
  • grammar: -

Author:  Luca Fausto Tocchio  on 2022-04-21  [id 2407]

(in reply to Report 2 on 2022-01-04)

We thank the referee for having carefully read our paper and for judging it suitable for publication, once some points are fixed. We report below an answer to the referee's comments and criticisms.

(1) It seems the authors have missed the fact that an extensive VMC study on the t-t' Hubbard model was already performed in [K. Ido, T. Ohgoe and M. Imada, PRB 97, 045138 (2018)] (however, without including backflow correlations). The authors cite this paper in their introduction (Ref. [35]), however, as a QMC calculation, and not as a VMC study. It would be important to compare their findings with these previous results in their discussion.

We are aware of the paper mentioned by the referee. We referred to it as QMC just because VMC is based upon a Monte Carlo sampling procedure of expectation values over quantum mechanical wave functions. We will refer to it as a VMC study in the revised version of the paper, in order to avoid any confusion. We have also included a discussion on this work in order to highlight differences and similarities. Indeed, apart from some differences in the definition of the wave function, our work and the one of Ref. 35 (Ref. 37 in the new version of the manuscript) are quite complementary. We considered different values of the Coulomb repulsion and the role of the sign of $t'/t$, that are not addressed in the other VMC work. Viceversa, the other VMC work considered different dopings, while we focused only on doping 1/8.

(2) The authors consider bond-centered stripes in their study. While there is a rather strong consensus that the lambda=8 stripe at 1/8 doping is bond-centered, this is not necessarily true for other stripe periods (or other dopings); in previous works there is rather a tendency that site-centered stripes are stabilized. Have the authors checked that site-centered stripes have a higher variational energy than bond-centered stripes? Or are the energies similar? It would be important to verify and discuss this point.

(3) It is not clear to me whether the authors have also considered odd values of lambda, since they only present results for even values. Have the authors checked the variational energies of odd lambda values and found that they are higher? Or was there a particular reason to restrict the study to even values? This point should be discussed/mentioned in the paper.

We thank the referee for highlighting these points. We have performed additional calculations, including odd values of $\lambda$ (leading to site-centered stripes) and even values of $\lambda$ with site-centered stripes. We found that only the case with $\lambda=5$ gives an energy improvement in a portion of the phase diagram for $t'/t<0$, while other odd values of $\lambda$ do not lead to a lowering of the variational energy. For even values of $\lambda$, site-centered stripes have the same energy (within the error bar) as bond-centered stripes, with the only exception of $\lambda=4$. In this case, site-centered stripes have an energy gain that is smaller than $10^{-3}$ with respect to bond-centered stripes. However, stripes with $\lambda=4$ never correspond to the best-energy wave function, because their energy is higher than the one of $\lambda=5$ stripes (see the revised phase diagram in the paper).

---

## Round 3 · Referee Report · Anonymous (Referee 2) · 2022-4-29

Report

The authors have substantially revised their manuscript and extended it with additional results. I find the new part on the comparison between site and bond centered stripes very interesting. The authors have also checked the variational energy of a pure PDW state. (In this context it would have been interesting to check also the energy of a PDW state coexisting with CDW and SDW orders, i.e. without setting \Delta_c= \Delta_s=0). In any case, the authors have addressed all points I raised in my previous report, and I can thus recommend this manuscript to be published in SciPost Physics.

There is one remaining small point which is not clear to me: on p5 the authors write: "Stripes with odd wavelength can only be site-centered". I do not see why they cannot be bond-centered (with a structure like odUdoodUdoodUdoodUdo… with U: large up spin, d: smaller down spin, o: a hole with a tiny up spin). I don't think this is energetically favored at 1/8 doping (maybe close in energy as in the lambda=6 case, or slightly higher as in the lambda=4 case), but it would be good to rephrase the statement (or explain why odd wavelength stripes cannot be bond-centered).

---

## Round 3 · Referee Report · Anonymous (Referee 1) · 2022-5-6

Report

I had a look at the replies and the modified paper.
The authors have responded to all concerns raised by the referee.
As I said in the previous report, the present work is one of the important pieces of recent intensive numerical investigation of the Hubbard model.
With additional calculations, the quality of the paper has improved.
I recommend that this paper be accepted for publication in SciPost.

---

## Round 3 · Author Response

Dear Editor,

we resubmit the manuscript, following the referees' comments and suggestions.

Your sincerely,
Luca F. Tocchio, on behalf of all the authors

---

## Round 3 · List of Changes

-) We included a brief description of the pair-density-wave scenario in the introduction, following a suggestion of the first referee.
-) We added in the introduction a comparison with the VMC work of the new Ref. 37, following a suggestion of the second referee.
-) We updated the variational wave function in the Method section, in order to describe also site-centered stripes.
-) We added a comment in the Method section, noticing that a superconductive state with the "antiphase" pairing, that can represent a pair density wave state, is never the optimal wave function.
-) We added a column to Table II, following a suggestion of the first referee.
-) We made the lines in the figures thicker, following a suggestion of the first referee.
-) We added a comment in the Results section, on the negligible difference between bond-centered and site-centered stripes when the wavelength is even, following a question of the second referee.
-) We updated the phase diagram, where now the (site-centered) stripe with wavelength 5 is present when t'/t<0. The inclusion of stripes with odd wavelength follows a question of the second referee.
-) We updated the phase diagram at U/t=12 and positive t'/t, where now the stripe with wavelength 12 is the optimal solution at t'/t=0.15 and t'/t=0.20.
-) We added two extra references, that is the new references 18 and 34.

---

## Editorial Decision

published